# Systematic Pan-Cancer Analysis Reveals X-C Motif Chemokine Receptor 1 as a Prognostic and Immunological Biomarker

**DOI:** 10.3390/genes14101961

**Published:** 2023-10-19

**Authors:** Likun Cui, Liye Zhu, Jie Chen, Chunzhen Li, Yizhi Yu, Sheng Xu

**Affiliations:** 1National Key Laboratory of Medical Immunology and Institute of Immunology, Naval Medical University, Shanghai 200433, China; poppy199609@163.com (L.C.); zhuly339@163.com (L.Z.); chenj12112021@163.com (J.C.); chunzhenli@smmu.edu.cn (C.L.); 2Shanghai Institute of Stem Cell Research and Clinical Translation, Shanghai 200120, China

**Keywords:** *XCR1*, pan-cancer, prognosis, immune infiltration

## Abstract

Chemokines and their receptors play an important role in immune monitoring and immune defense during tumor growth and metastasis. However, their prognostic roles in pan-cancer have not been elucidated. In this work, we screened all chemokine receptors in pan-cancer and discovered X-C Motif Chemokine Receptor 1 (*XCR1*) as a reliable immunological and prognostic biomarker in pan-cancer using bioinformation. The TCGA database served as the foundation for the primary research database analysis in this work. XCR1 was downregulated in tumors. Patients with reduced XCR1 showed worse prognoses and a concomitant decrease in immune cell infiltration (DCs and CD8^+^ T cells). According to a gene enrichment study, XCR1 enhanced immune system performance by promoting T-cell infiltration through the C-X-C Motif Chemokine Ligand 9 (CXCL9)- C-X-C Motif Chemokine Receptor 3 (CXCR3) axis. In addition, XCR1 is mainly expressed in infiltrated DCs and some malignant cells in tumor tissues. Our data revealed the important role of XCR1 in remodeling the tumor microenvironment and predicting the survival prognosis, which could also be used as a sensitive biomarker for tumor immunotherapy.

## 1. Introduction

Ligand–receptor interactions of chemokines and their receptors play a very important role in immune monitoring, inflammation, angiogenesis, tumor growth, and metastasis [1]. According to the configuration of the amino-terminal cysteine residues, chemokines are divided into four categories: CXC, CC, C, and CX (C stands for cysteine and X stands for non-cysteine amino acids) [2]. Chemokines act directly by binding to their receptors, which are seven transmembrane receptors conjugated with the G proteins. The binding of chemokines with their receptors causes conformational changes in the receptors, triggering intracellular signals which lead to directional cellular migration and other cellular events. The most important function of the molecules in the chemokine family is to promote the directional and non-directional migration of cells, induce molecular stagnation or adhesion, and also play a role in cell proliferation, survival, differentiation, cytokine production, degradation, and respiratory outbreaks [3]. Chemokine receptors are expressed differently in various kinds of immune cells and direct their migration to various locations, inducing either inflammatory sites or an antitumor immune environment.

Nowadays, cancer is still a leading public health problem all over the world. Chemokines and their receptors always play a very important role in tumor growth and proliferation. C-C Motif Chemokine Ligand 2 (CCL2) or C-X-C Motif Chemokine Ligand 8 (CXCL8) could promote tumor proliferation and invasion in an autocrine and paracrine manner. C-C Motif Chemokine Ligand 20 (CCL20)/C-C Motif Chemokine Receptor 6 (CCR6), C-C Motif Chemokine Ligand 25 (CCL25)/C-C Motif Chemokine Receptor 9 (CCR9), C-X-C Motif Chemokine Ligand 1 (CXCL1)/C-X-C Motif Chemokine Receptor 2 (CXCR2), CXCL8/C-X-C Motif Chemokine Receptor 1-2 (CXCR1-2), C-X-C Motif Chemokine Ligand 12 (CXCL12)/C-X-C Motif Chemokine Receptor 4 (CXCR4), and C-X3-C Motif Chemokine Ligand 1 (CX3CL1)/C-X3-C Motif Chemokine Receptor 1 (CX3CR1) are reported to promote tumor growth and inhibit apoptosis through the PI3K-Akt pathway [4]. C-C Motif Chemokine Receptor 4 (CCR4) is often overexpressed in several T-cell malignancies and is related to malignant proliferation [5,6]. C-C Motif Chemokine Receptor 7 (CCR7) can be pathologically expressed by tumor cells, which drives tumor growth and metastasis and increases their homing behavior, particularly towards lymphatic organs [7]. What is more, CCL2 and C-C Motif Chemokine Ligand 5 (CCL5) can promote cancer cell proliferation, survival, motility, epithelial–mesenchymal transition (EMT), and stemness [8]. CCL5 also promotes angiogenesis, which is associated with an increase in vascular endothelial growth factor (VEGF) expression in cancer cells and vascular endothelial cells via the activation of C-C Motif Chemokine Receptor 1 (CCR1) and C-C Motif Chemokine Receptor 5 (CCR5) [9,10,11]. However, there are also chemokines that suppress tumor growth. C-C Motif Chemokine Ligand 14 (CCL14) attenuates hepatocellular carcinoma (HCC) cell proliferation and promotes apoptosis through the Wnt/β-catenin signaling pathway [12,13].

Besides reacting to chemokines directly, cancer cells also acquire the abilities to initiatively disrupt the chemokine system from the early stages of cancer, tightly regulate the immune cell infiltration and tumor microenvironment to avoid oncoimmunity, and further affect malignant cell survival, metastasis, and spread in their microenvironment [14,15,16]. CCR1 overexpression has been described in several types of cancer and is associated with increased immunosuppressive cell infiltration and metastasis [17,18]. CCR5 is pathologically expressed in several types of cancer and upregulates oncogenic transformation, which could hijack the migration of immune cells and induce a homing behavior towards metastatic sites [19,20,21]. CXCL12, secreted by tumor and stromal cells, regulates plasmacytoid DC trafficking to tumors and integrin α5 and C-X-C Motif Chemokine Receptor 4 (CXCR4) expressed by plasmacytoid DCs and promotes tumor clearance through antigen presentation [22,23,24,25]. CCL2 expressed by tumors also correlates with the immune infiltration of tumor-associated macrophages (TAMs) in many tumors and is associated with poor tumor patient prognosis, including in breast cancer [26,27,28]. Therefore, the function of the chemokines and their receptors in tumors can be beneficial or harmful. Whether there are any chemokines or receptors that play a universal role in pan-cancer remains unclear.

X-C Motif Chemokine Receptor 1 (*XCR1*), the marker of cDC1s, not only plays an important role in the cross-presentation of APCs and T cells but also appears to impact the biological function of malignant cells due to its abnormal expression. In previous reports, XCR1 expressed in epithelial ovarian carcinoma (EOC) [29] and non-small-cell lung cancer [30] could promote their proliferation and migration. In contrast, it was also reported that XCR1 could inhibit cell proliferation, tumorigenesis, and tumor growth in breast cancer [31]. XCR1 was also reported to be expressed in epithelial cells in the normal oral mucosa and over-expressed and upregulated in both inflammatory oral diseases and oral cancers (including primary and metastatic cancers) [32].

However, the role and mechanism of XCR1 in tumor progression and the immune–cancer interaction are still unclear.

Here, we conducted a pan-cancer analysis of all chemokine receptors and found that only *XCR1* could serve as a prognostic marker in most cancers. We further analyzed the characteristics of *XCR1* in pan-cancer, including expression difference, correlation between expression levels and survival, genetic alterations, immune cell infiltration, and related signal pathways. We also investigated the functions and potential mechanisms of *XCR1* in clinical prognosis and pathogenesis. In summary, our data revealed the important role of *XCR1* in the prognosis of cancers, which could also be used as a sensitive biomarker for tumor immunotherapy.

## 2. Materials and Methods

### 2.1. Obtainment of Tissue Samples

Ten pairs of patients’ resected liver cancer tissues and their corresponding adjacent normal tissues were obtained from the Department of Thyroid and Breast Surgery, Changhai Hospital, with informed consent. These tissues were frozen and stored in liquid nitrogen until RNA extraction.

### 2.2. RNA Extraction and Quantitative Real-Time PCR

Total RNA was extracted from tissue samples using TRIzol reagent in accordance with the instructions of the manufacturer. Reverse transcription of RNA to cDNA used premix reverse transcription reagents (PrimeScript™ RT Master Mix, TAKARA(Shanghai, China), code No. RR036Q) following standard procedures. mRNA expression was quantified using quantitative real-time PCR with TB Green Premix Ex Taq (TAKARA(Shanghai, China), code No. RR820Q) and normalized with human GAPDH. Relative expression level was calculated using the 2^−ΔΔCt^ method. All primers were synthesized by Sangon Biotech Ltd. (Shanghai, China). The sequences of those primers are listed in Appendix A.

### 2.3. Gene Expression Difference Analysis

The R language (version 3.6.3) and R package “ggplot2” were used to analyze and visualize the RNAseq data of TCGA processed uniformly using the toiling process [33]. In this study, we analyzed the expression of *XCR1* across TCGA tumors in this way, and TCGA normal tissue data were matched as controls. RNAseq data in TPM (transcripts per million reads) format were analyzed and compared after log_2_ transformation with UCSC XENA (https://xenabrowser.net/datapages/, accessed on 3 April 2023).

### 2.4. Survival Prognosis Analysis and ROC Analysis

The TCGA datasets were used to investigate the correlation between *XCR1* expression and the prognosis of diverse cancers. The Kaplan–Meier plotter (https://kmplot.com/analysis/, accessed on 3 April 2023) was used to analyze the correlation between *XCR1* expression and the overall survival (OS) and disease-specific survival (DSS) with its groups “0–50” and “50–100”, respectively [34]. The R packages “pROC” and “ggplot2” were used to analyze and present the RNAseq data of TCGA. The area under the ROC curves (AUC) was calculated for the determination of diagnosis and prognosis [34].

### 2.5. Genetic Alteration Analysis

The cBioPortal web (https://www.cbioportal.org/, accessed on 3 April 2023) was used for genetic alteration analysis [35,36]. In total, 10,967 samples were chosen from 32 studies to acquire the genetic alteration characteristics of *XCR1*, which could provide information about the alteration frequency, mutation type, and CAN (copy number alteration) of all those tumors. We can also obtain the information on the overall survival and disease-specific survival with or without *XCR1* genetic alteration from this website.

### 2.6. Immune Infiltration Analysis

We used the TIMER2.0 (http://timer.cistrome.org/, accessed on 3 April 2023) database to analyze the immune infiltrates with different kinds of cancer types [36,37]. The database was used to evaluate the correlation between XCR1 expression and immune cell infiltration. The TIMER2.0 database provided a crucial assessment and integration of immune cells for RNA sequencing samples from TCGA, including B cells, natural killer cells, macrophages, neutrophils, dendritic cells, CD8^+^ T cells, and CD4^+^ T cells with their different subsets.

### 2.7. Gene Enrichment Analysis

The STRING website (https://string-db.org/, accessed on 3 April 2023) was used to acquire the XCR1-binding proteins and its protein–protein interaction network after functional enrichment analysis [36,38]. Then, we used GEPIA2.0 (http://gepia2.cancer-pku.cn/, accessed on 3 April 2023) to explore the correlated genes of *XCR1* in all kinds of cancer types. Finally, we obtained a list of *XCR1*-correlated genes, which was sorted by correlation coefficient. And then, the top 100 related genes were obtained. The KEGG website (https://www.kegg.jp/, accessed on 3 April 2023) was used for pathway analysis. Metascape (3.0) (http://metascape.org, accessed on 3 April 2023) was adopted to perform Gene Ontology (GO) analysis.

### 2.8. Public Single-Cell Database Analysis

The distribution of XCR1 expression in different cells was analyzed using the public single-cell database. Data were obtained from three datasets: NSCLC-007-02-1A, UCEC-024-01-1A, and NSCLC_GES127465, which separately show UMAP cluster plots and XCR1 expression statistics in different cell populations, utilizing the Cancer Single-cell Expression Map (https://ngdc.cncb.ac.cn/cancerscem/index, accessed on 3 April 2023) and TISCH2 (http: //tisch.comp-genomics.org/, accessed on 3 April 2023) websites.

### 2.9. Statistical Analysis

The statistical method used to investigate the expression difference in *XCR1* was a T-test with non-paired samples. The Wilcoxon rank-sum test was used to compare the expression levels of *XCR1* between tumor and the control tissue. The survival prognosis difference in *XCR1* was calculated using a Cox regression test. A Kaplan–Meier curve was drawn to evaluate the prognostic value of *XCR1*. The hazard risk (HR) of individual factors was estimated by measuring the HR with a 95% confidence interval (CI). Cox regression tests were used to analyze whether the grade of clinicopathological factors affects *XCR1* expression. Spearman’s correlation and the Wilcoxon rank-sum test were adopted to analyze the infiltration of immunocytes between the high- and low-expression groups of *XCR1*.

All *p*-values that were less than 0.05 were determined to be statistically significant. The statistical significance in following annotations is shown as follows: “*” means *p* < 0.05, “**” means *p* < 0.01, and “***” means *p* < 0.001.

## 3. Results

### 3.1. Pan-Cancer Screening of Chemokine Receptors Identifies Prognostic Value of XCR1

As the interaction of chemokines and their receptors determines immune cell infiltration in the TME, there is a great probability that chemokine family molecules play an important guiding role in the diagnosis and prognosis of cancers. To identify such molecules, we used the TCGA database and analyzed the relationship of patients’ survival with mRNA expression of each chemokine receptor in several common malignancies (Figure 1A). Among all the chemokine receptors, only *XCR1* was closely related with the hazard ratio in most types of cancers. By the way, there is no consistent phenomenon of X-C Motif Chemokine Ligand 1/2 (XCL1/2) in pan-cancer gene expression or patient survival prognosis (Appendix A).

Further Cox proportional hazards analysis showed that *XCR1* was a low-risk gene in SKCM (skin cutaneous melanoma, *p* < 0.001), LIHC (liver hepatocellular carcinoma, *p* < 0.001), HNSC (head and neck squamous cell carcinoma, *p* < 0.001), LUAD (lung adenocarcinoma, *p* = 0.003), ESAD (esophageal adenocarcinoma, *p* = 0.005), KIRC (kidney renal clear cell carcinoma, *p* = 0.012), BRCA (breast invasive carcinoma, *p* = 0.019), SARC (sarcoma, *p* = 0.021), CESC (cervical squamous cell carcinoma and endocervical adenocarcinoma, *p* = 0.034), and ESCA (esophageal carcinoma, *p* = 0.044) (Figure 1B). Furthermore, *XCR1* was not a significant high-risk gene in any type of cancer, suggesting the universal protective role of *XCR1* in human cancer.

### 3.2. Higher XCR1 Expression Is Related with Better Patient Survival

Then, we analyzed the relationship between *XCR1* expression and the overall survival (OS)/disease-specific survival (DSS) in tumor patients [39]. Kaplan–Meier survival analysis showed that a high expression of *XCR1* tends to be associated with a better prognosis of OS in all the above types of tumors (Figure 1C). Moreover, analysis of DSS data also revealed the association between low *XCR1* expression and poor prognosis in patients with SKCM (HR = 0.54 (0.41–0.7), *p* < 0.001), LIHC (HR = 0.42 (0.26–0.67), *p* < 0.001), HNSC (HR = 0.58 (0.41–0.82), *p* = 0.002), and CESC (HR = 0.45 (0.26–0.79), *p* = 0.005) (Figure 1D).

Then, diagnostic ROC (receiver operating characteristic) analysis was performed to assess the accuracy of prognostic efficacy with *XCR1*. The AUC value was greater than 0.7 in COAD (colon adenocarcinoma, AUC = 0.849), UCEC (uterine corpus endometrial carcinoma, AUC = 0.793), and KIRC (AUC = 0.728). It was also greater than 0.6 but less than 0.7 in LUSC (lung squamous cell, AUC = 0.696), LUAD (AUC = 0.671), THCA (thyroid carcinoma, AUC = 0.649), LIHC (AUC = 0.649), HNSC (AUC = 0.638), KICH (kidney chromophobe, AUC = 0.637), BLCA (bladder urothelial carcinoma, AUC = 0.609), and OSCC (oral squamous cell carcinoma, AUC = 0.602). These data revealed a good sensitivity and specificity of *XCR1* in the prognosis of survival in pan-cancer (Appendix A).

### 3.3. XCR1 Was Decreased in Human Cancer

We then analyzed the expression of *XCR1* among different tumors in the TCGA database with TIMER2.0 (Figure 2A). It has been noted that *XCR1* was expressed in a relatively low level both in tumor and adjacent normal tissue, according to the low TPM in all types of tumors. However, compared with adjacent normal tissue, an even lower expression of *XCR1* was observed in most cancers (Kruskal–Wallis test *p* < 0.05), such as LIHC, LUAD, LUSC, READ (rectum adenocarcinoma), THCA, UCEC, and HNSC (Figure 2B). In particular, in KIRC, *XCR1* expression was higher than it was in nearby normal tissues. The change in *XCR1* expression level in tumors indicates that it may be a protective factor related to the occurrence and development of tumors.

### 3.4. Low Expression of XCR1 in Advanced Cancer Stages

Next, we investigated the clinical features of *XCR1* according to the pathologic stages of the patients in the TCGA project. We found that in most common cancer types, *XCR1* expression levels were significantly lower in almost all stages, including in COAD, LIHC, LUAD, LUSC, and THCA (Figure 2C). In contrast, its expression was elevated in all stages in KIRC, suggesting a specific effect of *XCR1* in KIRC. Furthermore, different degrees of differences in *XCR1* expression were found between different tumor advance stages in all these types of cancer: T stage (the size and spread of the primary tumor, ranging from T0 (no detectable primary tumor) to T4 (significant tumor size and spread)), N stage (the extent of lymph node involvement, with N0 denoting no lymph node involvement and N1–N3 representing increasing degrees of lymph node involvement), and M stage (the presence or absence of distant metastasis, categorized as M0 (no distant metastasis) or M1 (distant metastasis present)) [40] (Figure 3A–J). Among them, the expression of *XCR1* in normal samples of several tumor patients in T, N, and M stages was significantly different from that in advanced tumor stages, such as LUAD, LUSC, KIRC, COAD, LIHC, READ, and THCA (Figure 3B,D–H,J). In particular, the expression difference in *XCR1* in LUSC can be used to distinguish the T2 and T3 stages of tumors and to indicate the stage of the T stage (Figure 3D). The differential expression of *XCR1* in different tumor stages suggests that *XCR1* can be used as a clinical staging marker to assist in cancer diagnosis and treatment. Among above cancers, we further explored the multifactorial Cox regression analysis of *XCR1*. The results showed that *XCR1* was associated with a good survival prognosis (OS/DSS/PFI) and positively correlated with the T stage of clinical staging. However, there was no significant difference in other factors, such as gender, age, BMI, etc. (Appendix A). This suggests that the low expression of *XCR1* is closely related to the occurrence, poor development, and spread of cancer cells.

### 3.5. Mutation Features of XCR1 in Pan-Cancer

In order to further elucidate the mutational characteristics and biological functions of *XCR1* in tumor occurrence and progression, we investigated the genetic alteration status of *XCR1* in pan-cancer based on the cBioPortal database [36,41]. Missense and truncating are the most important types of *XCR1* gene mutations (Appendix A). *XCR1* mutations are less frequent and have no direct impact on the overall survival of the patients (Appendix A). In addition, the missense and truncating mutations of *XCR1* were the main two types of genetic alterations: Y14Tfs*33 truncating alteration was the high-frequency mutation, which was detected in three cases of STAD (stomach adenocarcinoma), one case of COAD, and one case of UCEC; A33T/S alteration was an important site mutation, which was detected in cases of COAD, STAD, OV (ovarian serous cystadenocarcinoma), and UCEC (Appendix A). Furthermore, the genetic alteration analysis of the alteration frequency with *XCR1* was >4%, and the primary types were deletion and missense, such as in DLBC (Lymphoid Neoplasm Diffuse Large B-cell Lymphoma), KIRC, and SKCM, etc. (Appendix A).

### 3.6. XCR1 Expression Correlation with Immune Infiltration of CD8^+^ T Cells

Although the above results support the prognostic implications of *XCR1* in different cancers, its potential role warranted additional investigations. Immune cells play a crucial role in the immune microenvironment and can affect the prognosis of patients with cancer. We then investigated the relationship between *XCR1* expression and immune cell infiltration levels in diverse cancer types. The *XCR1* expression level was significantly correlated with immune cell infiltration in most types of cancer, especially BRCA, COAD, KIRC, KIRP (kidney renal papillary cell carcinoma), PAAD (pancreatic adenocarcinoma), PRAD (prostate adenocarcinoma), READ, SARC, SKCM, STAD, and TGCTs (testicular germ cell tumors). The TIMER database indicated that *XCR1* was mainly positively correlated with CD4^+^ T cells, CD8^+^ T cells, neutrophils, macrophages, and DCs in these tumors (Figure 4A, left). The expression of *XCR1* was confirmed to be significantly associated with CD8^+^ T-cell and DC infiltration using the XCELL algorithm (Figure 4A, right). No significant correlation was found between *XCR1* and the infiltration of NK cells.

We continued to analyze the effect of *XCR1* expression on CD8^+^ T-cell infiltration in different tumors and plotted the corresponding scatter plots with the TIMER database. The expression of *XCR1* was positively correlated with CD8^+^ T cells in most cancers, such as PAAD (Rho = 0.620), STAD (Rho = 0.556), KIRP (Rho = 0.411), TGCT (Rho = 0.406), etc. (Figure 4B). It was concluded that in pan-cancer, *XCR1* expression can promote the infiltration of DCs and CD8^+^ T cells, which indicates a better survival expectation and potential sensitivity to immunotherapy.

### 3.7. CXCL9-CXCR3 Axis Was Involved in Immune Infiltration by XCR1

To further study the molecular mechanism of *XCR1* in tumorigenesis and development, we decided to screen out XCR1-binding proteins for protein–protein interaction network analysis. Using the STRING online tool, we obtained 10 XCR1-binding proteins, including members which were supported by experimental evidence or prediction (Figure 5A). The directly acting molecules verified were XCL1, XCL2, C-C Motif Chemokine Ligand 13 (CCL3), C-C Motif Chemokine Ligand 8 (CCL8), C-C Motif Chemokine Ligand 19 (CCL19), C-C Motif Chemokine Ligand 24 (CCL24), and CXCL9. And then we also conducted gene co-expression analyses to explore the relationships between *XCR1* and associated genes.

To compare the common genes between the related genes and interacting genes of *XCR1*, an intersection analysis of the top 100 related genes and top 20 interacting proteins was conducted, shown by an interactive Venn diagram. We obtained only one common molecule, named CXCL9 (Figure 5B). Then, we analyzed the molecular correlation between XCR1 and CXCL9 in different tumors. It was found that XCR1 and CXCL9 have a significant positive correlation in a variety of tumors, including THCA, SKCM, SARC, KIRC, TGCT, BRCA, PCPG (pheochromocytoma and paraganglioma), and KIRP (Figure 5C). What is known is that there is a strong positive correlation between CXCL9 and XCR1 (Figure 5C). SKCM, BRCA, and LIHC tumor patients whose XCR1 was positively associated with their survival prognosis (Figure 1C) were selected for survival analysis (OS) and presentation. The results showed that *CXCL9* was associated with favorable survival in SKCM (HR = 0.58, *p* < 0.001) and BRCA (HR = 0.66, *p* = 0.013) but not in LIHC (HR = 0.84, *p* = 0.328) (Appendix A).

Other related genes, like C-type lectin domain containing 9A (*CLEC9A*) and basic leucine zipper ATF-Like transcription factor 3 (*BATF3*), were characteristic genes expressed in *XCR1*^+^ DCs [42,43] or CD8^+^ T cells [44]. It is known that CXCR3 is the receptor for CXCL9, CXCL10, and CXCL11 and mediates the proliferation, survival, and angiogenic activity through a heterotrimeric G-protein signaling pathway. Thus, it is highly likely that XCR1 promotes T-cell and DC infiltration and antitumor effects by affecting the CXCR3-CXCL9 axis. We also found strong positive correlations between XCR1 and CXCR3 in a variety of tumors, including SKCM, UVM (uveal melanoma), SARC, MESO (mesothelioma), PCPG, BRCA, CHOL (cholangiocarcinoma), and OV (Figure 5D). In conclusion, all these data strongly support the conclusion that *XCR1* expression is significantly correlated with DC and CD8^+^ T-cell infiltration in the TME. It is reported that classical dendritic cells (cDC1) expressing can secrete chemokines such as CXCL9 and recruit T cells and granulocytes after receiving signals [45]. These results suggest a strong link between the XCR1-XCL1/2 pathway and the CXCR3-CXCL9 pathway.

In addition, we performed Kyoto Encyclopedia of Genes and Genomes (KEGG) pathway analysis and Gene Ontology (GO) enrichment analysis based on SKCM and LIHC (Figure 5E,F). The KEGG enrichment analysis indicated that XCR1 plays a role in tumors through the “cytokine-cytokine receptor interaction” and “chemokine signaling pathway”, which suggests that XCR1 may play a role in recruiting and recognizing immune cells and transmitting signals through itself as a chemokine receptor. In addition, we also found that the gene set “cell adhesion molecules” was also enriched in LIHC, which also confirms previous reports that the expression of XCR1 is associated with tumor cell metastasis and spread in the later stages of cancers [30]. The GO enrichment analysis suggested that these genes were mainly related to “immune response-activating cell surface receptor signaling pathway”, “humoral immune response”, “lymphocyte mediated immunity”, “adaptive immune response based on somatic recombination of immune receptors built from immunoglobulin superfamily domains”, “immunoglobin complex”, and “antigen binding”, strongly indicating its key role in tumor immunity in the microenvironment.

### 3.8. XCR1 Mainly Expressed in DCs and Malignant Cells

To discriminate the cell types that express XCR1, we analyzed the distribution of XCR1 in different types of cancer. We analyzed the distribution of XCR1 in the immune cells from three different datasets: NSCLC-007-02-1A, UCEC-024-01-1A, and NSCLC_GES127465 (Figure 6A–C). Cell clustering and the gene expression profile confirmed that XCR1 was mainly expressed in a subset of DCs and malignant cells. The tissue samples from liver cancer patients we collected found that *XCR1* and *CXCR3* mRNA expression levels in tumor tissues were significantly lower than in adjacent tissues, while *CXCL9* shows no significant difference (Figure 6D–F). These results may indicate that *XCR1* not only plays an important role in recruiting immune cell infiltration in tumor microenvironments but may also be related to the biological functions of tumor cells.

## 4. Discussion

The tumor microenvironment (TME) leads to the occurrence of immune infiltration and affects the development of tumors. Immune cells in the TME infiltrate secrete inflammatory mediators, forming an inflammatory microenvironment with a high degree of heterogeneity. Among the immune cells, CD8^+^ T cells are important immune cells that kill pathogens and tumors, which are regulated by DC cross-presentation. Both migratory DCs and resident DCs can be further subdivided into cDC1s and cDC2s [46,47,48,49]. cDC1s, which depend on the transcription factors interferon regulatory factor 8 ( *IRF-8*)*,* inhibitor of DNA binding 2 (*Id2*), and *Batf3* for their development [50,51,52], selectively express the C-type lectin receptor *CLEC9A* (*DNGR-1*) [53,54] and the chemokine receptor *XCR1* [51,55,56]. *XCR1* is the signature marker of cDC1s, which is generally considered to have a low expression in normal tissues. XCR1 expressed by cross-presenting DCs plays a role, as it is known that XCR1 engagement by its ligand, XCL1, promotes the interaction between cross-presenting DC and CD8^+^ T cells to enhance the tumor immunity of immune system [56,57]. In this work, we used the TCGA database to perform relatively complete data mining with common tumors; it was found that the decrease in *XCR1* expression corresponded to the poor survival of the tumor patients from the screening of all chemokine receptor families, which suggested the value of *XCR1* in the survival diagnosis of tumor patients.

Compared with normal tissues, the expression of *XCR1* in pan-carcinoma is generally lower, except in KIRC, although it is not clear whether the *XCR1* decline is due to a decrease in *XCR1* expression by tumor cells or a decrease in cDC1 invasion. However, it can be verified that patients may have a poor survival prognosis when *XCR1* expression declines in the tumor tissue. The opposite survival prognosis in KIRC may correspond to the high expression in tumors compared with the adjacent normal tissue. ROC analysis also proved that *XCR1* could be applied to current precancerous diagnoses. More tumor diagnostic indicators could help to better classify tumor patients and facilitate different-group treatments. We also found that the expression of *XCR1* in tumor tissues accompanies a positive correlation in immune cell infiltration (especially T cells, neutrophils, and dendritic cells). As a result, the constitutive expression of *XCR1* in tumor tissues can be used as a clinical diagnostic and prognostic indicator of pan-cancer.

We found that the expression of *XCR1* in the N stage in different tumors is significantly lower than that of its para-cancerous tissues, indicating that there is an obvious possibility of lymphatic metastasis in *XCR1* lowest tumors. Furthermore, the obvious lower expression in the M stage means that tumor cells with a lower expression of *XCR1* have a stronger distal metastasis ability. This was consistent with a previous report in which malignant oral mucosal epithelial cells were prone to metastasis when *XCR1* expression decreased [32]. In the clinical stage analysis of pan-cancer, it was found that the low expression of *XCR1* means the progression of the tumor. And in patients with advanced cancer, the reduction in or loss of *XCR1* was always accompanied by lymph node metastasis and the distal metastasis of tumor cells.

Immune cell (except for suppressive cells, such as Tregs, MDSCs, TAMs, etc.) infiltration could help to monitor and kill tumor cells and prolong the survival of cancer patients. The combination of XCR1-XCL1/2 helps to promote the recruitment of T cells, and our GSEA enrichment of the CXCL9-CXCR3 axis is involved in the XCR1-mediated recruitment of T cells [58]. CXCR3 is known to have a key role in T-cell trafficking to inflammatory sites and is required for immune cell trafficking to tumor tissues [59,60]. CXCL9 is secreted by CD103^+^ DCs and binds to CXCR3 on T cells and then promotes T-cell recruitment [61,62]. Recruited and activated T cells continue to secrete XCL1 [1], which binds to XCR1 and further enhances the cross-presentation of DCs to CD8^+^ T cells. Therefore, the interaction of the above two sets of chemokines suggests that the expression of XCR1 forms a positive feedback loop in tumor immunity. It is also consistent with the positive correlation we found between the expression of *XCR1* and the infiltration of CD8^+^ T cells and other immune cells in tumor tissues. Overall, our pan-cancer analyses of *XCR1* have elaborated the landscape of *XCR1* expression with the clinical prognosis and immune cell infiltration, which provide a predictive biomarker in pan-cancer, especially in LIHC, KIRC, LUAD, HNSC, and COAD. At the same time, the possible molecular mechanism of XCR1-mediated immune invasion in tumor tissues suggested that the infiltration of immune cells is promoted by a positive loop consisting of the XCR1/XCL1 axis and CXCR3/CXCL9 axis, leading to the understanding of XCR1′s important role in tumor immunity. Furthermore, we have to admit that not all tumor types are consistent with our observations. KIRC expression and the survival prognosis of UVM are relatively special cases in this paper because they contradict our finding that XCR1 plays a positive role against tumors. In accordance with the principle of being realistic, we have chosen to present these two typical examples of the opposite phenomenon but with significant differences. It would also be helpful to judge and rule out special cases in diagnosis and treatment if *XCR1* was used as a biomarker for cancer detection.

## 5. Conclusions

In conclusion, we used public databases to screen a sensitive biomarker of the chemokine receptor family in multiple carcinomas in this work. We found that *XCR1* was related to a good prognosis in pan-carcinoma and showed a positive correlation between *XCR1* expression and immune cell invasion after further data mining. Gene enrichment analysis suggested that the XCR1-XCL1/2 axis enhances immune cell, especially CD8^+^ T cell, infiltration by promoting the binding of CXCL9 to CXCR3. These results provide a predictive biomarker and an inclusive comprehension of *XCR1* expression in pan-cancer.

## Figures and Tables

**Figure 1 genes-14-01961-f001:**
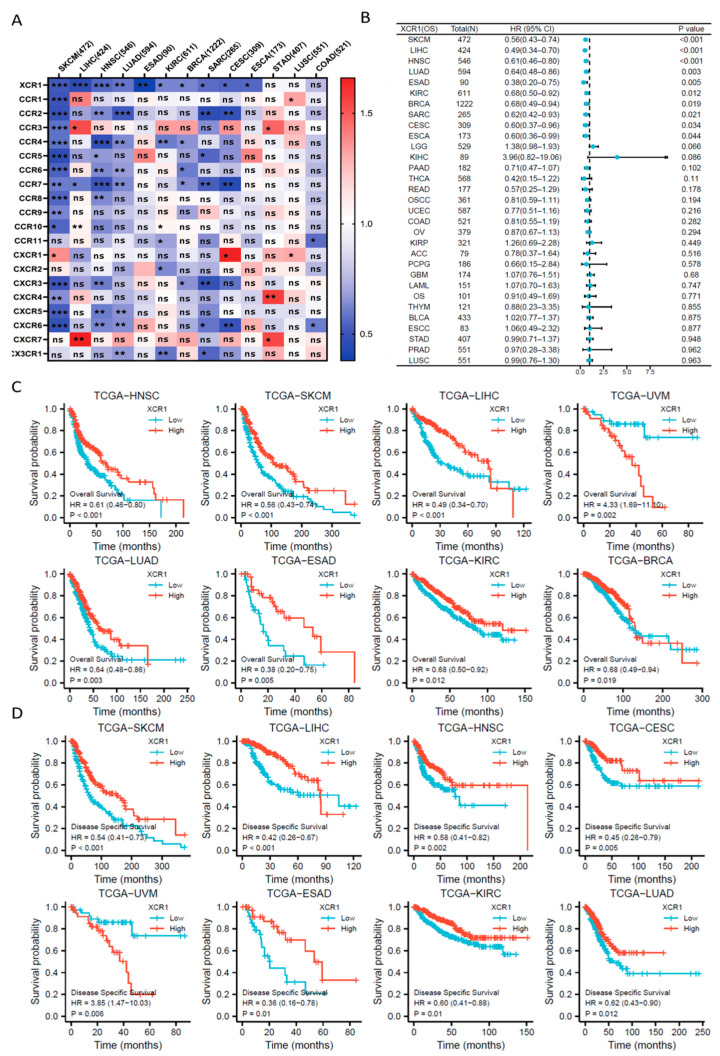
Screening prognostic relevance of *XCR1* in pan-carcinoma. (**A**) Prognostic correlation (overall survival) heatmap of all chemokine receptors in most common human tumors. “*” means *p* < 0.05, “**” means *p* < 0.01, “***” means *p* < 0.001, and “ns” means not significant. (**B**) Forest plots of *XCR1* expression in pan-cancer. HR: hazard ratio. (**C**) Overall survival analysis of *XCR1* among several types of tumors (HNSC, SKCM, LIHC, UVM, LUAD, ESAD, KIRC, and BRCA) in the TCGA dataset. (**D**) Disease-specific survival analysis of *XCR1* genes among several types of tumors (SKCM, LIHC, HNSC, CESC, UVM, ESAD, KIRC, and LUAD) in the TCGA dataset.

**Figure 2 genes-14-01961-f002:**
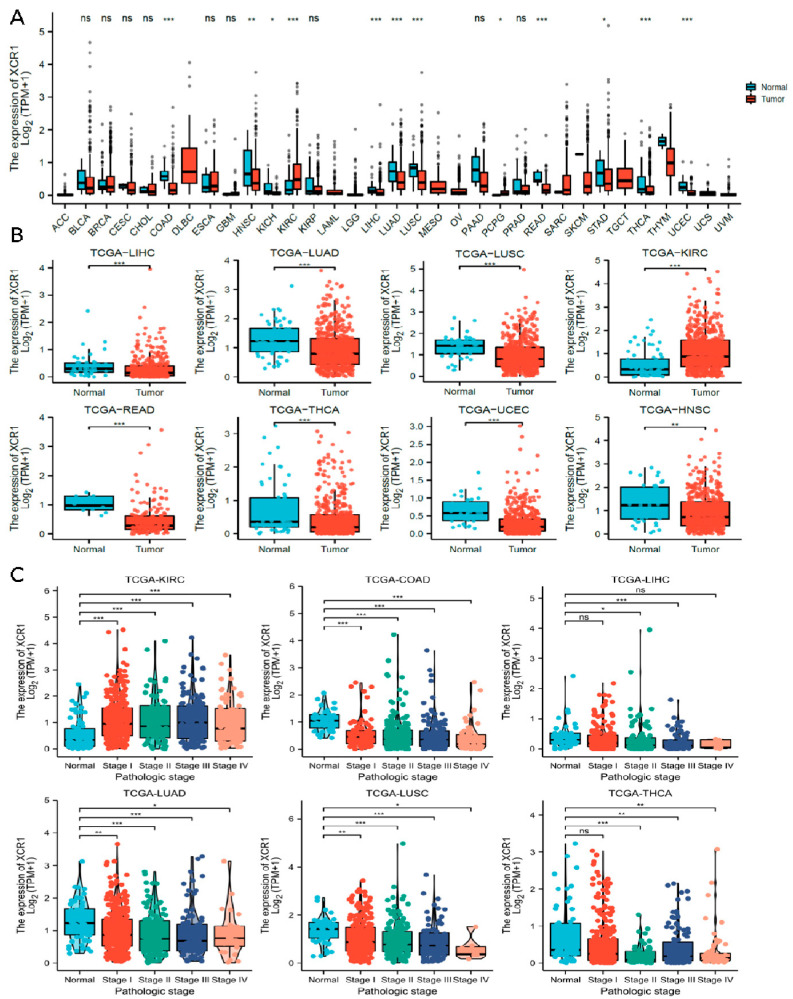
The expression and pathologic-stage relevance of *XCR1* in different cancers. (**A**) Human *XCR1* expression levels in pan-cancer with its para-cancerous normal tissues from TCGA database. (**B**) Scatter plots of *XCR1* expression in LIHC, LUAD, LUSC, KIRC, READ, THCA, UCEC, and HNSC; para-cancerous tissues were included as controls. (**C**) The expression of *XCR1* according to the pathologic stage of the tumor patients in TCGA. “*” means *p* < 0.05, “**” means *p* < 0.01, “***” means *p* < 0.001, and “ns” means not significant.

**Figure 3 genes-14-01961-f003:**
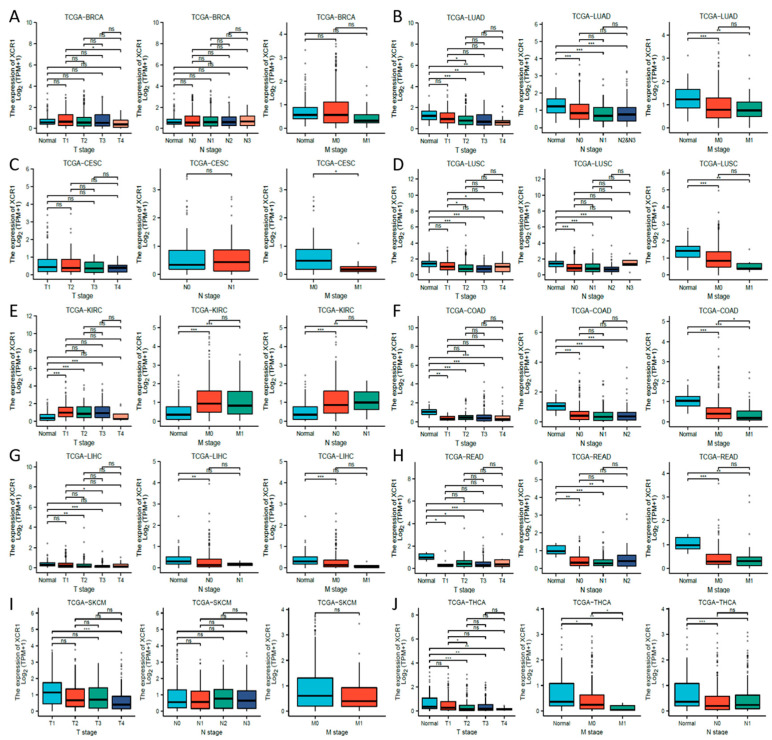
*XCR1* expression in different tumor pathologic stages. (**A**–**J**) The expression of *XCR1* according to the pathologic stage of the patient in the TCGA cancer types. Each picture includes the T stage (the size of the primary tumor), N stage (lymph node metastases), and M stage (distal metastatic conditions) from left to right. “*” means *p* < 0.05, “**” means *p* < 0.01, “***” means *p* < 0.001, and “ns” means not significant.

**Figure 4 genes-14-01961-f004:**
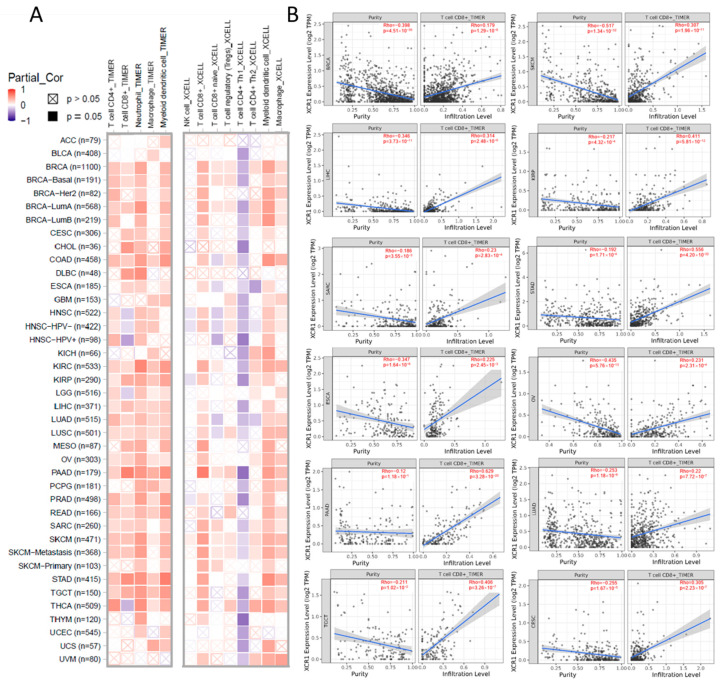
Correlation analysis between *XCR1* expression and CD8^+^ T-cell infiltration. (**A**) Correlation analysis between *XCR1* expression and immune infiltration. The analysis in the left picture was calculated with the TIMER algorithm and the right with the XCELL algorithm. (**B**) The heatmap and scatter plot represent the relationship between CD8^+^ T-cell infiltration and *XCR1* gene expression of all types of cancer in the TCGA project.

**Figure 5 genes-14-01961-f005:**
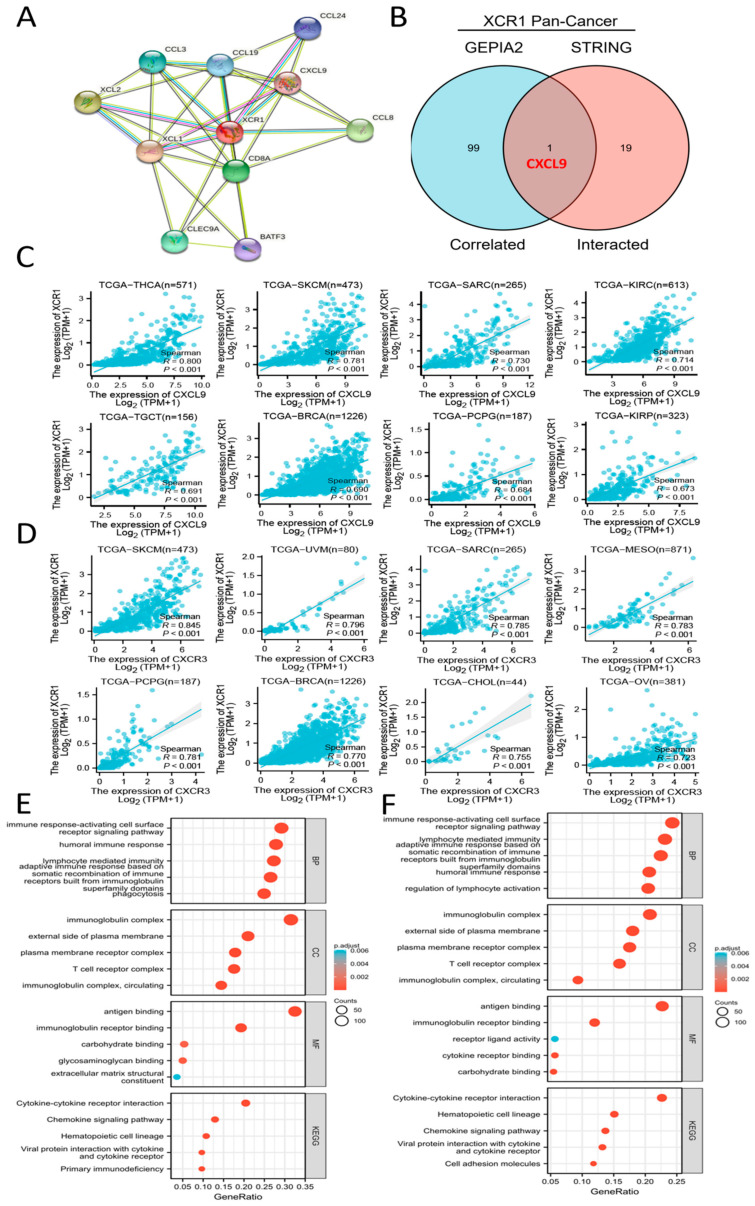
*XCR1*-related gene enrichment analysis. (**A**) *XCR1* interaction network analysis using STRING database. (**B**) Intersection analysis of *XCR1*-correlated genes and *XCR1*-interacting genes. (**C**) Correlation between *XCR1* and CXCL9 in different cancers. (**D**) Correlation between *XCR1* and *CXCR3* in different cancers. (**E**) GO/KEGG enrichment analysis based on the *XCR1*-correlated and -interacting genes in SKCM. (**F**) GO/KEGG enrichment analysis based on the *XCR1*-correlated and -interacting genes in LIHC.

**Figure 6 genes-14-01961-f006:**
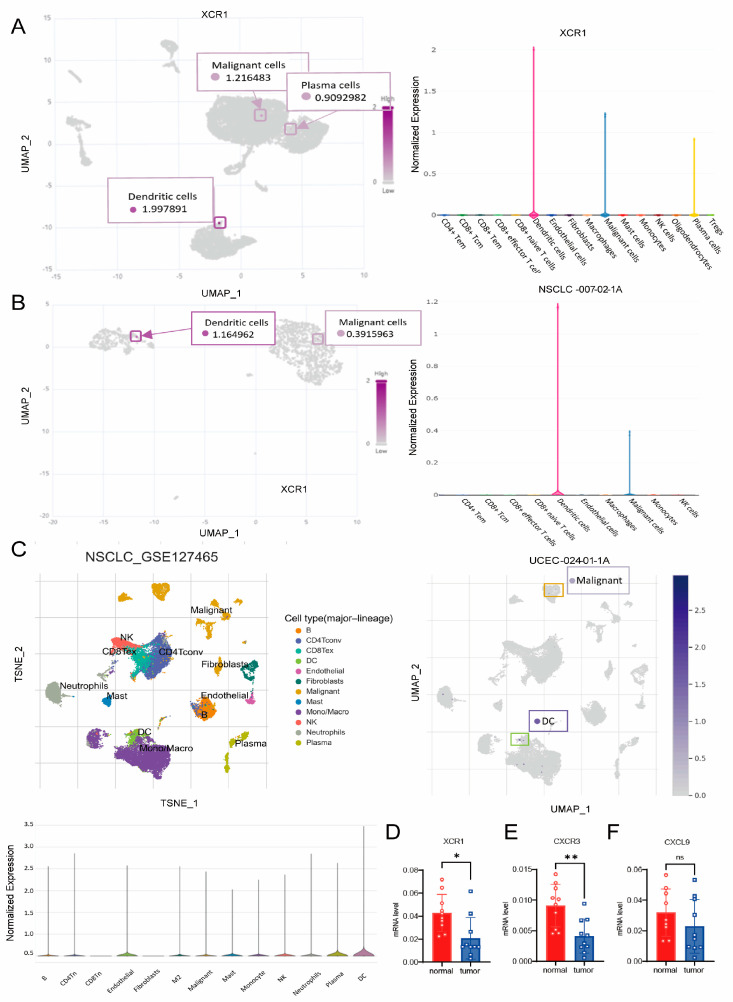
scRNA sequencing reveal *XCR1* expression in DCs and malignancy cells. (**A**–**C**) The distribution and expression of *XCR1* in the immune cells and tumor cells from three different datasets: (**A**) NSCLC-007-02-1A, (**B**) UCEC-024-01-1A, and (**C**) NSCLC_GES127465. (**D**–**F**) mRNA expression level of *XCR1*, CXCR3, and CXCL9 in tissue samples from liver cancer patients, *n* = 10. “*” means *p* < 0.05, “**” means *p* < 0.01, and “ns” means not significant.

## Data Availability

Publicly available datasets were analyzed in this study. These data can be found as follows: UCSC XENA (https://xenabrowser.net/datapages/, accessed on 3 April 2023), TIMER database (http://timer.cistrome.org/, accessed on 3 April 2023), GEPIA2.0 (http://gepia2.cancer-pku.cn/, accessed on 3 April 2023), the Kaplan–Meier plotter (http://kmplot.com/analysis/, accessed on 3 April 2023), the cBioPortal web (https://www.cbioportal.org/, accessed on 3 April 2023), the STRING website (https://string-db.org/, accessed on 3 April 2023), and three datasets, NSCLC-007-02-1A, UCEC-024-01-1A, and NSCLC_GES127465, from Cancer Single-cell Expression Map (https://ngdc.cncb.ac.cn/cancerscem/index, accessed on 3 April 2023) and TISCH2 (http: //tisch.comp-genomics.org/, accessed on 3 April 2023).

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
