# Peer review of "Systematic Pan-Cancer Analysis Reveals X-C Motif Chemokine Receptor 1 as a Prognostic and Immunological Biomarker"

_genes, 2023, doi:10.3390/genes14101961_

Round 1

Reviewer 1 Report

The authors provide evidence for a protective role for chemokine receptor XCR1 in multiple cancers. The finding seems interesting since higher XCR1 shows survival benefits in patients with different types of cancers. The authors have used the TCGA database, thus providing good power to the study. Some of the comments are below.

The authors showed XCR1 but did not include its traditional cytokine ligands. Did the author look into the expression of XCL1/2 in the tumor? Was survival based on microarray data? Did RNA seq data show similar results for it? The timer dataset probably needs more explanation as to the expression of xcr1 in various populations of immune cells. The single-cell datasets are not very well described in the method sections.

Additionally, did the author look into TCGA single-cell datasets if available? The connection between CXCl9 and XCR1 is interesting, and the author could cite the reference from String if it was direct experimental evidence. Also, the coexpression of CXCL9 and XCR1 might be a just correlation since CXCL9 could still bind to its cognate receptors, as the authors looked into CXCR3. However, the section on CXCL9-CXR3 possibly needs more experimental evidence. The authors could look into whether CXCL9 and XCR1 high tumors have any survival benefits or how CXCL9 correlated with survival.

The authors could also rephrase their statements at 300-301. The confusion could be because of how xcr1 is explored in the text. Could the author show where XCR1 is supposed to have a tumor suppressor role? Also, the qPCR lists two primers; which one was used, and what control gene primer was used?. How do the authors explain the high XCR1 in their patients? Do these patients have high CXCL9 as well? The other option will be to check other cancer datasets to see if XCR1 has a protective role.

Reviewer 2 Report

This study underscores the significance of XCR1 as a valuable immunological and prognostic biomarker across various cancer types. Notably, downregulation of XCR1 in tumors is linked to unfavorable patient outcomes and decreased immune cell infiltration. XCR1's ability to bolster the immune response by facilitating T cell infiltration via the CXCL9-CXCR3 axis is a noteworthy finding, and it is primarily expressed in infiltrated dendritic cells and certain malignant cells. Consequently, XCR1 emerges as a promising marker for predicting survival outcomes and guiding immunotherapy strategies, shedding new light on its potential.

However, there are a couple of issues to consider:

  1. The statement, "Furthermore, significant differences were found between different tumor advance stages in all these types of cancer: T stage (the size of the primary tumor), N stage (lymph node metastases), M stage (distal metastatic conditions) (Figure 3A-J)," is inaccurate. Some of the figures in Figure 3 do not hold statistical significance.

  1. The statement, "The results showed that XCR1 was associated with a good survival prognosis (OS/DSS/PFI) and positively correlated with the T stage," contradicts the findings in Figure 3. XCR1 expression levels were significantly lower in some cancer types at the T stage.

These inconsistencies should be addressed and clarified in the study for a more accurate interpretation of the results.

Minor editing of English language required

Round 2

Reviewer 1 Report

The authors have responded comprehensively.